# Functional and Radiographic Results of Arthroscopy-Assisted Lateral Open-Wedge Distal Femur Osteotomy for Lateral Compartment Osteoarthritis with Valgus Knee

**DOI:** 10.3390/jcm12010176

**Published:** 2022-12-26

**Authors:** Ruei-Shyuan Chien, Cheng-Pang Yang, Chun-Ran Chaung, Chin-Shan Ho, Yi-Sheng Chan

**Affiliations:** 1Department of Orthopedic Surgery, Division of Sports Medicine, Chang Gung Memorial Hospital, College of Medicine, Chang Gung University, Taoyuan 333, Taiwan; 2Comprehensive Sports Medicine Center, Chang Gung Memorial Hospital, Taoyaun 333, Taiwan; 3Graduate Institute of Sports Science, National Taiwan Sport University, Taoyuan 333, Taiwan; 4Bone and Joint Research Center, Chang Gung Memorial Hospital, Taoyuan 333, Taiwan; 5Department of Orthopedic Surgery, Chang Gung Memorial Hospital, Keelung 204, Taiwan

**Keywords:** lateral compartment osteoarthritis, valgus knee malalignment, lateral open-wedge distal femur osteotomy, knee arthroscopy

## Abstract

Treating lower extremity malalignment-related knee osteoarthritis, especially valgus alignment, is a challenge. A high revision rate was observed with patients who underwent unicompartmental knee arthroplasty, so distal femur osteotomy has regained its popularity. This research aimed to evaluate the radiographic and functional outcomes of arthroscopy-assisted lateral open-wedge distal femur osteotomy (LOWDFO) for patients with lateral compartment osteoarthritis and valgus knees with a minimum follow-up of 2 years. Our study retrospectively included isolated lateral osteoarthritis (Outerbridge grade 3 and grade 4) of the knee related to valgus alignment and a young age (<65 y/o) with the demand for a high-impact activity event. Preoperative and postoperative radiographic and functional outcomes were evaluated. Significant pre-operative and postoperative mechanical correction was observed with mechanical axis deviation (preop/postop: −28.77 ± 12.98/−9.45 ± 7.36, *p* < 0.001), hip-knee angle (preop/postop: 7.64 ± 3.62/2.68 ± 2.04, *p* < 0.001), and mechanical lateral distal femoral angle (mLDFA, preop/postop: 10.9 ± 4.14/5.66 ± 3.71, *p* < 0.001). The International Knee Documentation Committee (IKDC) score also showed improvement after the operation (preop/postop: 57.36 ± 11.98/79.02 ± 4.58, *p* = 0.002). In conclusion, lateral open-wedge distal femur osteotomy is effective in treating patients with lateral compartment osteoarthritis and valgus knees with a low complication rate and excellent outcome.

## 1. Introduction

Lower extremity malalignment-related knee osteoarthritis (OA), including valgus and varus alignment, in young patients remains a critical problem for orthopedic surgeons. Varus alignment was shown to be associated with a four-fold increase in the odds of medial progression of OA, whereas valgus alignment was associated with a nearly five-fold increase in the odds of lateral progression [1]. An increasing number of studies have revealed that varus-alignment-related knee OA can be treated successfully by high tibial osteotomy (HTO), which aims to shift the mechanical axis from the diseased compartment to the healthy compartment. HTO showed even more prominent results in greater survival and clinical outcomes when combined with arthroscopy, articular cartilage surgery, or meniscal allograft transplantation [2]. However, few studies have focused on the effectiveness of dis femoral osteotomy (DFO), which aims to treat lateral OA by correcting the mechanical axis to the medial compartment.

Lateral OA can also be treated by unicompartmental knee arthroplasty (UKA). However, lateral UKA had a high revision rate due to the dislocation of the insert and a low 5-year survival rate [3,4]. As a result, DFO is regaining its popularity for young and active patients who are seeking joint preservation. The potential advantage of DFO, similar to HTO, includes native joint preservation, a return to unrestricted high-impact activity after the union of the osteotomy, and a possible delay of future arthroplasty [5]. A concern regarding this procedure is that a variant outcome of DFO was reported with a major complication of the nonunion and plate irritation [6,7]. Few studies have focused on the results when DFO is combined with concomitant procedures, such as arthroscopy [8]. Hence, our study aimed to evaluate whether DFO combined with arthroscopy could improve the accuracy of the mechanical axis correction and clinical outcome.

## 2. Materials and Methods

We retrospectively assessed patients who underwent distal femoral osteotomy from January 2019 to April 2020 by a single surgeon. All patients had lateral knee pain over the joint line, and preoperative lower limb split scanography, knee X-ray, and MR images were compatible with symptomatic lateral compartment overload and valgus knee malalignment. Surgical intervention was indicated for failure of nonoperative management, including activity modification, anti-inflammatory medications, and physical therapy for more than 3 months. The indication for distal femoral osteotomy included isolated lateral osteoarthritis (Outerbridge grade 3 and 4) of the knee related to valgus alignment and a relatively young age (<65 y/o) with the demand for a high-impact activity event. Patients with rupture of the cruciate ligament, moderate-to-advanced medial compartment cartilage wear, septic knee arthritis, or rheumatoid arthritis were excluded.

### 2.1. Preoperative Assessment

Preoperative assessment included the anterior–posterior view and lateral view of the knee for OA grading. Split scanography of the lower extremities was also taken for knee alignment evaluation. We documented the angle by mechanical axis deviation (MAD), hip–knee angle, joint line congruency angle (JLCA), and mechanical lateral distal femoral angle (mLDFA) (Figure 1). The correction angle was defined as the hip–knee angle measured by low extremity split scanography. Every patient underwent MRI of the knee for evaluation of cartilage wear, bone edema, or soft tissue injury, including meniscus or ligament tears (Figure 2).

### 2.2. Surgical Procedure

We performed our technique similar to the method reported by Puddu et al. [9] by using LISS plate and Synthes^®^ plate. Knee arthroscopy was performed initially. Three portals were inserted from three directions: the anteromedial, anterolateral, and superomedial (outflow portal) directions. Meniscus and cartilage damage was checked with arthroscopy. Microfracture or meniscus repair was performed as indicated (Figure 3). The operative technique used throughout this study was the lateral opening wedge technique. First, lateral incision of iliotibial band and dissection between vastus lateralis and biceps femoris were performed with direct dissection to bone. The plate was then applied with one osteotomy from 2–3 cm proximal to the lateral epicondyle toward the medial epicondyle. All steps were performed under a fluoroscopic image guide. The open wedge correction angle was made as planned, and then a Synthes LISS^®^ locking plate was applied. The osteotomy site was filled with Stimulan^®^ and femoral head structural allograft. Under fluoroscopic imaging, the mechanical axis was aimed so that it passed through the middle of the knee to reach anatomical correction (Figure 4).

### 2.3. Postoperative Care

For the first 6 weeks after the procedure, patients were restricted to partial weight-bearing activity. Knee range-of-motion movements were encouraged as tolerated. After 6 weeks, the patient could start a quad-strengthening and progressive weight-bearing rehabilitation program. A gradual return to activity and sport-specific training was not allowed until 3 months after the procedure. Every patient was followed up for at least 2 years. Split scanography of the lower extremities was performed for postoperative alignment evaluation (Figure 5). The International Knee Documentation Committee (IKDC) score was used to assess the functional outcome of DFO. Preoperative versus postoperative outcome scores and tibiofemoral alignment were assessed using Wilcoxon Signed Rank test. All statistical analyses used 2-tailed tests and were considered to reach statistical significance at *p* < 0.05.

## 3. Results

From Jan. 2019 to Nov. 2020, 16 patients met the indications for DFO, including an age below 65 years, high activity demand, and isolated lateral osteoarthritis (Outerbridge grade 3 and 4) according to lower limb X-ray and valgus alignment. Patient characteristics are shown in Table 1.

The mean follow-up time was 2.5 years. Under arthroscopy, 11 patients were considered to have a modified Outerbridge classification grade 4 in the lateral compartment, whereas the remaining 5 patients were considered to be classified as grade 3. A total of 13 patients underwent lateral meniscus tear repair, including 12 patients with all inside techniques and 1 patient with inside-out techniques. All patients received microfractures over the exposed subchondral bone. The average operation time was 173.53 ± 38.53 min, with an average intraoperative blood loss of 68.33 ± 50.52 mL. The average hospital stay was 3.87 ± 0.62 days. Preoperative and postoperative radiographic and functional evaluations and changes are shown in Table 2.

Statistically significant changes were noted in MAD, hip–knee angle, and mLDFA. A decrease in JLCA was observed postoperatively (Preop/Postop: 2.35 ± 1.18/1.75 ± 1.23), but the data did not reach statistical significance (*p* = 0.179). During the 2-year follow-up, the MAD, hip–knee angle, and mLDFA showed a decrease in value compared to those of postop. When compared to preop, all radiological parameters still remained improved with statistical significance at the 2-year follow-up.

The mean union time of the lateral open wedge DFO was 3.7 ± 2.12 months (1.9–9.5 months). Only one patient had a stitch abscess noted at 2.5 months after surgery. There was no hinge fracture or delayed union (union time above 12 months). No patients underwent plate removal, conversion to TKA, or other knee surgery during the 2-year follow-up.

Clinically, the patient showed an improvement in daily function, assessed by the IKDC score, with statistical significance (preop/postop: 57.36 ± 11.98/79.02 ± 4.58, *p* = 0.002). During the 2-year follow-up, an improved IKDC score was noted compared to that of the postoperative period (postop/2-year: 79.02 ± 4.58/92.45 ± 3.11, *p* = 0.007).

## 4. Discussion

Our study included 16 patients who underwent lateral open-wedge DFO with arthroscopy procedures. The results showed a significant improvement postoperatively in the mechanical axis deviation, hip–knee angle, and mLDFA. Our results also showed an increase in the IKDC score, from 57.36 ± 11.98 to 79.02 ± 4.58, which is similar to the findings of previous studies. According to A. Saithna. et al., whose research included a survey of 21 lateral open wedge osteotomies, the cumulative survivorship determined by using conversion to arthroplasty as an endpoint was 79% at 5 years [10]. A. Zarrouk et al. showed a promising result, with an 8-year survival rate of 91% (confidence interval, 69–100%) in lateral open-wedge DFO [11]. For other common postoperative surgeries, a study showed that 47% of patients had metalwork removed due to localized discomfort or tenderness [10]. Dewilde et al. revealed that 4 out of 19 patients who had an opening-wedge osteotomy underwent hardware removal, 1 patient underwent fracture fixation, and 2 patients were converted to TKA. The survivorship at 7 years with revision surgery or conversion to TKA was 82% [12].

The surgical indication for DFO varies from surgeon to surgeon. Few studies reported the actual indication for DFO, and most of the studies gave vague inclusion criteria, including a mechanical axis deviated by > 15 mm laterally or a mLDFA above one standard deviation from normal [13]. Another study included patients who were young and active (<65 years old), had valgus malalignment (<20 degrees), and had isolated lateral compartment osteoarthritis [14]. Jacobi included patients that met the following criteria: mechanical axis deviation above 10 mm (normal value 9.7 ± 6.8 mm) combined with a reduced mLDFA (normal value 88°); age was not a criterion to exclude individuals from the study [15]. Our indication for lateral open-wedge DFO included isolated lateral osteoarthritis of the knee related to valgus alignment and a relatively young age (<65 y/o) with the demand for high-impact activity events. By the indication above, we could avoid including patients with osteoporosis, resulting in a higher risk of complications and preserving natural joint connective tissue to reach a feasible functional outcome.

There are two different approaches for valgus alignment osteotomy, including medial closed-wedge (MCWDFO) and lateral open-wedge osteotomy (LOWDFO). There is still debate over which approach is better. The advantages of performing an MCWDFO are direct bone apposition, construct stability, reliable bone healing, no need for bone graft, and an inherent stability allowing for immediate partial weight bearing. The disadvantages include an increased risk for neurovascular injury caused by the proximity to the popliteal vessels and less ability to fine-tune intraoperative correction [16,17]. On the other hand, the advantages of performing a LOWDFO are a familiar surgical approach, better control of the intraoperative correction, and access to the lateral aspect of the knee where the disorder is most commonly found, enabling concomitant procedures [18]. The disadvantages include a need for bone grafts, a longer time needed for bony healing, an increased risk of malunion or nonunion, potential hardware irritation, and non–weight-bearing ambulation for 8 weeks postoperatively. Jacobi et al. stated that LOWDFO was associated with a slower healing period, with 7 out of 14 patients remaining crutch-dependent 3 months after the operation [15]. However, we did not observe a similar outcome in our study, which may be because we applied allografts at the osteotomy site to every patient. James D. Wylie conducted a systemic review [19] that included 16 studies with 372 osteotomies. There were more conversions to arthroplasty rates in participants in the MCWDFO group (22%), whereas only a 12% conversion to arthroplasty rate was seen in the LOWDFO group. However, the mean follow-up of participants in the MCWDFO group was over 10 years, whereas only 78 months of the mean follow-up period were seen in participants in the LOWDFO group. More reoperations to remove the plate due to irritation were seen in participants in the LOWDFO group. Due to a high heterogeneity within the included studies, there was no conclusion regarding which approach was better, and further study is needed.

It is well stated that arthroscopy should be regularly performed in DFO surgery. The American Academy of Orthopedic Surgeons suggested routinely performing arthroscopy of intra-articular pathology before DFO. The advantage of arthroscopy-assisted surgery includes identifying medial compartment injury, which may be contraindicated for DFO, and bicompartment knee injury may benefit more from arthroscopy alone if patients have mechanical symptoms, or patients may benefit from other arthroplasty options if nonsurgical treatment is unsuccessful [5]. Arthroscopy also allows for the best assessment of the chondral surfaces in the unaffected compartments, although these arthroscopic findings of bicompartmental or tricompartmental arthritis have not been found to be predictive of outcome in the mid-term follow-up. Future study is needed for a better predictive value [20]. Arthroscopy could also identify surgically treatable intraarticular pathology, such as the removal of a meniscal flap or anterior tibial osteophyte, while adding little to the morbidity of the procedure [9]. In a retrospective study, 21 patients underwent DFO with concomitant lateral meniscal allograft transplantation. The results revealed a high rate of return to sports at an average of 16.9 months postoperatively, as well as a significant decrease in VAS pain scores (from preoperative 5.7 to postoperative 2.6, *p* = 0.02) [21]. Evidence of improving the postoperative outcome by combining corrective osteotomy and a knee arthroscopy procedure is mostly reported in HTO [22]. O-Sung Lee et al. performed a microfracture on patients with full-thickness articular cartilage lesion. A total of 67.8% patients had cartilage regeneration during a second-look operation of 2 years [23]. In our study, 13 patients received arthroscopic meniscus repair, and all 16 patients received arthroscopic debridement with a microfracture.

To our knowledge, this is the first case series focusing on the functional outcome of arthroscopy-assisted DFO in Taiwan. However, there are still limitations in our study. First, only 16 patients were included in our case series, which is a small number. Second, the mean follow-up period of our study was short, so future evaluation, including plate removal due to plate irritation, second-look knee arthroscopy, and rate of conversion to TKA, is needed.

## 5. Conclusions

Arthroscopy-assisted lateral open-wedge distal femur osteotomy is effective in treating patients with lateral compartment osteoarthritis with valgus knees, as it has a low complication rate and excellent outcome at a minimum of 2 years of follow-up.

## Figures and Tables

**Figure 1 jcm-12-00176-f001:**
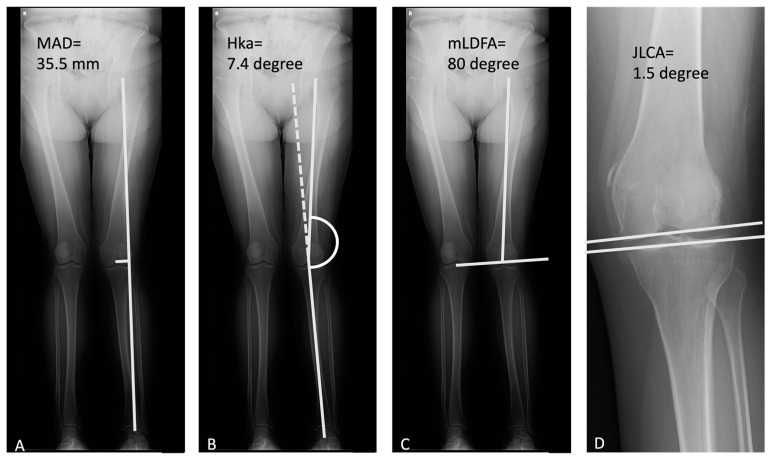
Radiographic parameters: (**A**) mechanical axis deviation, MAD (35.5 mm), (**B**) hip–knee angle, HKa (7.4 degree), (**C**) mechanical lateral distal femoral angle, mLDFA (80 degree), (**D**) joint line congruency angle, JLCA (1.5 degree).

**Figure 2 jcm-12-00176-f002:**
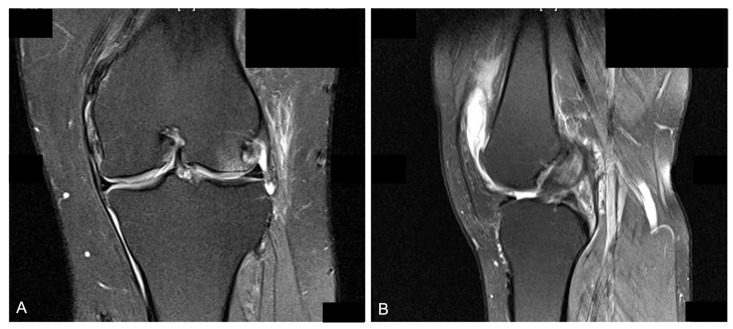
Preop MRI: (**A**) under T2-weighted image, coronal view showed lateral meniscus tear, lateral compartment cartilage disruption, and subchondral bone edema; (**B**) under T2-weighted image, sagittal view showed intact anterior cruciate ligament.

**Figure 3 jcm-12-00176-f003:**
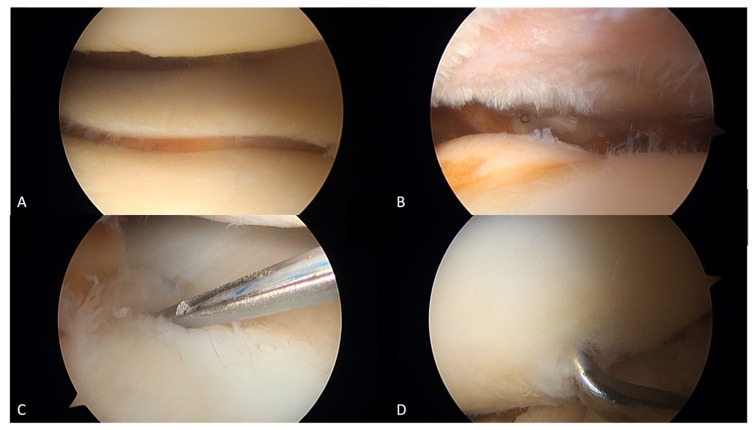
Arthroscopic image: (**A**) intact medial meniscus and good quality of cartilage, (**B**) lateral meniscus tear and lateral compartment OA change, Outerbridge grade 4, (**C**) lateral meniscus repair with all-inside technique, (**D**) microfracture of femoral condyle.

**Figure 4 jcm-12-00176-f004:**
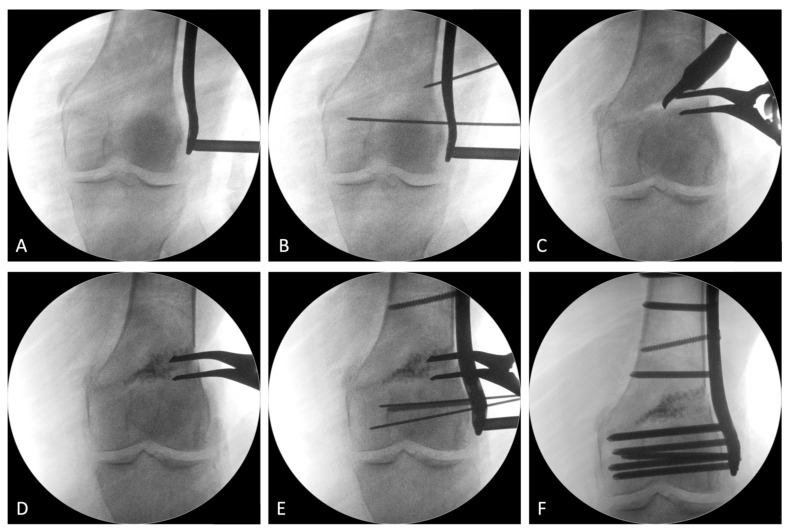
The fluoroscopy images show the operative steps. (**A**) The placement of Synthes LISS locking plate. (**B**) Pin was inserted from lateral femoral condyle toward medial femoral condyle. (**C**) Open wedge was performed by lamina spreader. (**D**) Osteotomy site was filled with Stimulan^®^ and femoral head structural allograft. (**E**) Unicortical locking screws were placed in the distal segment first, followed by a bicortical oblique nonlocking screw to compress the osteotomy gap. (**F**) Final image for corrected alignment.

**Figure 5 jcm-12-00176-f005:**
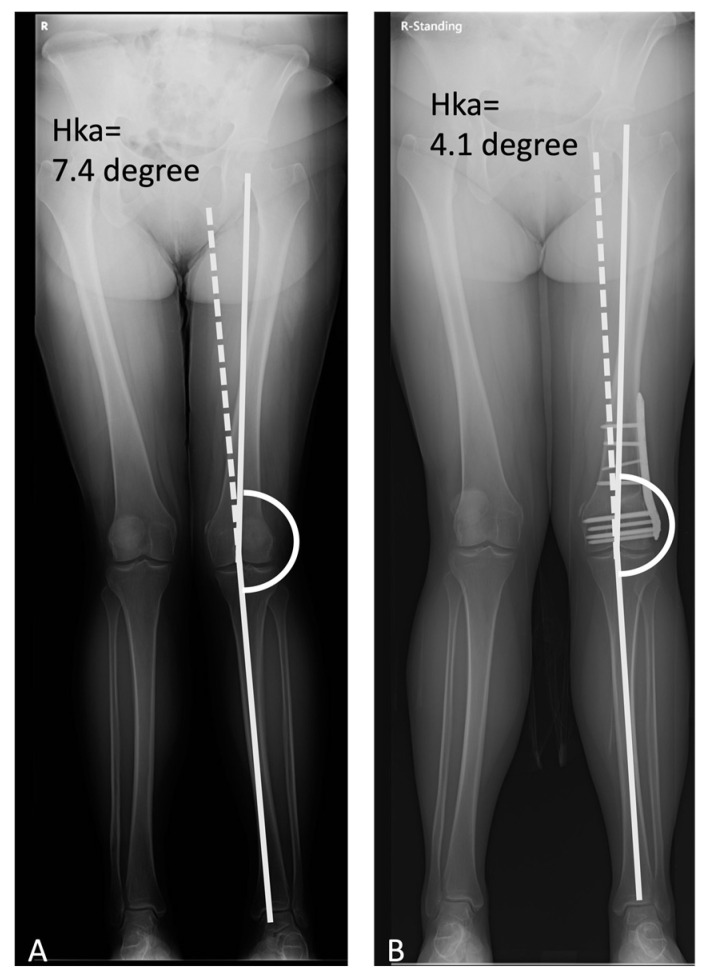
(**A**) Preop, Hka = 7.4 degree and (**B**) postop, Hka = 4.1 degree, split scanography.

**Table 1 jcm-12-00176-t001:** Demographic data.

Numbers	16
Age (years old)	44.81 ± 13.82 (24–65)
Sex	
Male	25% (4/16)
Female	75% (12/16)
Side	
Left	50% (8/16)
Right	50% (8/16)
BMI (kg/m^2^)	25.65 ± 4.29 (18–37)

**Table 2 jcm-12-00176-t002:** Mechanical axis and functional result of preoperative and postoperative change.

	Preop	Postop	Postop 2-Year
MAD (mm)	−28.77 ± 12.98	−9.45 ± 7.36 *	−14.6 ± 8.7 *
Hip-knee angle (degree)	7.64 ± 3.62	2.58 ± 1.93 *	3.2 ± 1.63 *
JLCA (degree)	2.35 ± 1.48	1.75 ± 1.61	1.3 ± 0.83 *
mLDFA (degree)	83.51 ± 3.48	92.01 ± 3.41 *	90.2 ± 2.77 *
IKDC score	57.36 ± 11.98	79.02 ± 4.58 *	92.45 ± 3.11 *

* with statistical significance of *p* < 0.05 compared to preop data. MAD: mechanical axis deviation. JCLA: joint line convergence angle. mLDFA: mechanical lateral distal femoral angle.

## Data Availability

Not applicable.

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
