# Peer review of "Functional and Radiographic Results of Arthroscopy-Assisted Lateral Open-Wedge Distal Femur Osteotomy for Lateral Compartment Osteoarthritis with Valgus Knee"

_jcm, 2022, doi:10.3390/jcm12010176_

Round 1

Reviewer 1 Report

1) There must be needed on the result of radiological parameters on 2-year follow-up. Furthermore, longitudinal comparative analysis should be performed.

2) Detailed illustration of inclusion/exclusion criteria should be needed (including smoking history et al…)

3) Total number of patients is too small.

4) Operative outcomes (such as operative time, hospitalization et al.) should be presented in Results section

5) For DFO, What is the merit of A/S assisted lateral open wedge DFO compared to other conventional methods? In the present description, it does not feel that there is a distinctions advantage. It is necessary to describe in detail the advantages of being different from the existing method in the disruption.

Author Response

1) There must be needed on the result of radiological parameters on 2-year follow-up. Furthermore, longitudinal comparative analysis should be performed.

  • We appreciate the reviewer’s insight. No malunion or loss of correction during 2 years follow up. The limitation for this study is that no comparative group analysis has been done and future study is expected for further evaluation.

2) Detailed illustration of inclusion/exclusion criteria should be needed (including smoking history et al…)

  • We thank the reviewer for the comment. We have added a more thoroughly inclusion and exclusion criteria to the manuscript.

3) Total number of patients is too small.

  • We thank the reviewer for the comment. We have managed to recruit more cases for analysis with the minimum follow up for 2 years. Future study would be expected for larger patient number and longer follow up period.

4) Operative outcomes (such as operative time, hospitalization et al.) should be presented in Results section

  • We thank the reviewer for the precious comment. More detailed operative outcomes have been added to the manuscript. Average operation time was 173.53 ± 38.53 minutes with average intraoperative blood loss of 68.33 ± 50.52ml. Average hospital stay was 3.87 ± 0.62 days.

5) For DFO, What is the merit of A/S assisted lateral open wedge DFO compared to other conventional methods? In the present description, it does not feel that there is a distinctions advantage. It is necessary to describe in detail the advantages of being different from the existing method in the disruption.

  • If performed distal femur osteotomy(DFO) without knee arthroscopic procedure, intra-articular lesion would be neglected(meniscus tear, foreign body, chondral wear). However, there were few articles discussed about DFO combined with arthroscopic procedure. We performed a 2-year follow up case series study and excellent radiological and functional outcome were achieved. In the future, the comparison of DFO with and w/o arthroscope could be arranged just like HTO.

Reviewer 2 Report

Dear Authors, the topic is interesting because it is about the application of Arthroscopy-Assisted  Lateral Open Wedge Distal Femur Osteotomy for Lateral Compartment Osteoarthritis with Valgus Knee

As regards the Introduction

I suggest that it could be useful to insert a brief chapter about OA, where should be cited a recent article about it:

 - Bizzoca D, Vicenti G, Solarino G, et al. Gut microbiota and osteoarthritis: a deep insight into a new vision of the disease. Journal of Biological Regulators and Homeostatic Agents. 2020 Sep-Oct;34(5 Suppl. 1):51-55. IORS Special Issue on Orthopedics. PMID: 33739005.

The approval of the ETCO Committee for the recruitment of patients to the study is lacking. Is there a misprint or there isn’it?

As regards the Discussion,

I suggest that it could be useful to insert a brief chapter where there are articles about  

- the role of arthroscopy In grade 3 and grade 4 of Outerbridge in order to understand why it was inserted as an inclusion criteria

Author Response

I suggest that it could be useful to insert a brief chapter about OA, where should be cited a recent article about it: 

 - Bizzoca D, Vicenti G, Solarino G, et al. Gut microbiota and osteoarthritis: a deep insight into a new vision of the disease. Journal of Biological Regulators and Homeostatic Agents. 2020 Sep-Oct;34(5 Suppl. 1):51-55. IORS Special Issue on Orthopedics. PMID: 33739005.

  • We appreciate the reviewer’s insight. The article mentioned above was mainly about how gut microbiotacould affect osteoarthritis and current scientific evidence. Our study presented here did not include the above concept and was mainly discussed shifting mechanical axis by distal femur osteotomy. We would kindly include the aspect in our future study related to osteoarthritis.

The approval of the ETCO Committee for the recruitment of patients to the study is lacking. Is there a misprint or there isn’t it?

  • The study was conducted in accordance with the Declaration of Helsinki, and approved by the Institutional Review Board of Chang Gung Medical Foundation (protocol code 202201146B0 and was approved on 2022/08/01).

As regards the Discussion, 

I suggest that it could be useful to insert a brief chapter where there are articles about  

- the role of arthroscopy In grade 3 and grade 4 of Outerbridge in order to understand why it was inserted as an inclusion criteria

  • We thank the reviewer’s for the comment. Evidence of improving postoperative outcome by combining corrective osteotomy and knee arthroscopy procedure is mostly reported in high tibial osteotomy. We have added the previous evidence in the manuscript.

Round 2

Reviewer 1 Report

Dear authors,

The flow of this study has been progressed but it has many limitation. For limitation, it should be illustrated in the manuscript.

1. First of all, the idea of this study has sufficient academic value.

2. Statistical analysis should be illustrated and need to be improved in Material and Methods section. Total number of patients was 16. Is it reasonable a choices of paired-t-test in this study ?

3. The data at 2-year follow-up was still not presented in this study. It is reasonable to conclude "as it has a low complication rate and excellent outcome at a minimum of 2 years of follow-up." without 2-year follow-up clinical data (IKDC score) and radiological parameters. 

Author Response

  1. First of all, the idea of this study has sufficient academic value.

We kindly thank the reviewer for the recognition.

  1. Statistical analysis should be illustrated and need to be improved in Material and Methods section. Total number of patients was 16. Is it reasonable a choices of paired-t-test in this study ?

We appreciate the reviewer’s insight. Due to the total number of patients was 16, we decided to use Wilcoxon Signed Rank test for data analysis. Re-assessed data was demonstrated in the revised version of manuscript and did not affect the final conclusion.

  1. The data at 2-year follow-up was still not presented in this study. It is reasonable to conclude "as it has a low complication rate and excellent outcome at a minimum of 2 years of follow-up." without 2-year follow-up clinical data (IKDC score) and radiological parameters. 

We thank the reviewer for the comment. We have added clinical and radiological parameters at 2 years follow up in our revised manuscript. Future study, including larger patient number and long follow-up period, would be expected for the efficacy of this procedure.  

Reviewer 2 Report

no further comment

Author Response

We appreciate for previous precious insight and comments!